# Virus Eradication and Synthetic Biology: Changes with SARS-CoV-2?

**DOI:** 10.3390/v13040569

**Published:** 2021-03-28

**Authors:** Jean-Nicolas Tournier, Joseph Kononchik

**Affiliations:** 1Microbiology and Infectious Diseases Department, Institut de Recherche Biomédicale des Armées, 91220 Brétigny-sur-Orge, France; jean-nicolas.tournier@intradef.gouv.fr; 2CNRS UMR-3569, Innovative Vaccine Laboratory, Virology Department, Institut Pasteur, 75015 Paris, France; 3Ecole du Val-de-Grâce, 75005 Paris, France; 4US Army Medical Research Institute of Chemical Defense (USAMRICD), 8350 Ricketts Point Rd., Aberdeen Proving Ground, MD 21010, USA; 5Toxicology and Chemical Risk Department, Institut de Recherche Biomédicale des Armées, 91220 Brétigny-sur-Orge, France

**Keywords:** vaccine, synthetic biology, virus, eradication, outbreak, public, opinion, polio, infectious disease

## Abstract

The eradication of infectious diseases has been achieved only once in history, in 1980, with smallpox. Since 1988, significant effort has been made to eliminate poliomyelitis viruses, but eradication is still just out of reach. As the goal of viral disease eradication approaches, the ability to recreate historically eradicated viruses using synthetic biology has the potential to jeopardize the long-term sustainability of eradication. However, the emergence of the severe acute respiratory syndrome-coronavirus (SARS-CoV)-2 pandemic has highlighted our ability to swiftly and resolutely respond to a potential outbreak. This virus has been synthetized faster than any other in the past and is resulting in vaccines before most attenuated candidates reach clinical trials. Here, synthetic biology has the opportunity to demonstrate its truest potential to the public and solidify a footing in the world of vaccines.

## 1. Introduction

In 1980, the eradication of smallpox was celebrated by the World Health Organization (WHO) as a great achievement and triumph [1]. In combination with the success of antibiotics, it foreshadowed a new era of triumph over infectious diseases. Indeed, smallpox eradication was viewed as the first global public health victory aimed at curing the world of endemic diseases. Among the top diseases targeted are malaria, polio, and measles.

Now, after three decades of effort, the malaria program has been discontinued [2], and polio eradication is incomplete. The conclusion to the WHO’s plan for global poliovirus (PV) eradication in 2020 seems to constantly be moving especially as the older oral vaccine provokes outbreaks of circulating vaccine-derived poliovirus (cVDPV) [3]. By taking advantage of new technology, such as those found in synthetic biology, the ability to quickly make targeted, effective, and low side-effect vaccines may be the catalyst needed to eradicate it and other infectious diseases; and an opportunity to demonstrate the safety and efficacy of newer technology.

The emergence of severe acute respiratory syndrome-coronavirus-2 (SARS-CoV-2) has resulted in the worst pandemic in a century. With synthetic biology and reverse genetics, we have targeted many diseases with vaccines, including seasonal influenza [4]. What gets noticed, however, is the publication of the synthesis of poliovirus in 2002 [5] and of horsepoxvirus from commercially purchased sequences [6]. These headlines raised new concerns on their potential reemergence and synthetic biology’s “dual-use” [7]. Artificially synthesized viruses can intrinsically threaten global health efforts as a direct biosecurity threat, or through the spread of previously tacit knowledge. Despite these problems, SARS-CoV-2 has shown that, with synthesis of the virus within a week, it can challenge the traditional pathway of vaccine generation [8]. This article discusses balancing the perspective of the good and the bad of this technology in a global fight against viral infections.

## 2. The Challenges of Virus Eradication

The eradication of infectious diseases is one of the oldest human dreams. Since its establishment after World War II, the WHO has launched four global programs for the eradication of infectious diseases targeted at malaria, smallpox, poliomyelitis and Dracunculiasis (Guinea worm disease, a parasitic infection caused by *Dracuncula medinensis*). In 1955, the first WHO program was devoted to malaria. It used a vector transmission targeting strategy through the wide use of the residual insecticide dichloro-diphenyl-trichloroethane (DDT) [2]. The DDT strategy was mainly driven by results from the US army that achieved malaria control in several locations during the World War II [9]. Unfortunately, soon after the program started, the rapid development of pesticide-resistant mosquitoes led to the re-emergence of malaria. Eventually, the malaria program was discontinued in 1969. Here, we will focus on human viral diseases, though, as we have demonstrated that they can be eradicated more realistically using our current vaccine technology.

The WHO implements and defines its own standards and efficiency objectives [10]. Eradication of an agent is the highest level. It signifies that the global incidence of the disease has been reduced to zero by the physical elimination of the agent. Eradication, however, can be distinguished from extinction; meaning its physical elimination from the surface of the earth (including laboratories). There is no known example of viral extinction, since the smallpox virus is kept in repositories in the USA and Russia, and the 1918 Spanish flu virus, which spontaneously disappeared after 1919, was recreated in 2005 [11].

It took over 150 years from Edward Jenner’s first experimental vaccination with cowpox in 1796 to the beginning of the smallpox global eradication program in 1958 [1] (Figure 1). The eradication campaign lasted almost two decades and ended in 1977 with the last case of smallpox. During the second half of the 20th century, many virus vaccines were developed; boosted by major technological advances in cell and virus cultures [12]. Global eradication campaigns, unfortunately, do not merely benefit from technological advancements, but rely on international politics and robust financing, which can complicate a campaign before it even begins [13].

In addition to its eradication campaigns, in 1974 the WHO launched the expanded program on immunization (EPI). The initial goals of the EPI were to ensure that every child received protection against the six major childhood diseases (i.e., tuberculosis (TB), polio, diphtheria, pertussis, tetanus and measles) by age one as well as give tetanus toxoid vaccinations to women to protect them and their newborns. In 1978, the International Conference on Primary Health Care (PHC) at Alma-Ata, Kazakhstan (formerly USSR) expressed the need for urgent action by all governments, all health and development workers, and the world community to protect and promote the health of all people. It was the first international declaration underlining the importance of PHC. This approach has since been accepted by numerous countries as the key to achieving the goal of “Health-For-All”. The Alma-Ata declaration of 1978 emerged as a major milestone of the 20th century in the field of public health. Vaccine campaigns were one of the primary tools for reaching the major goals of the Alma Ata declaration. In 2000, the EPI was replaced by the Global Alliance for Vaccines and Immunization (GAVI), a public–private global health partnership with the goal of increasing access to immunization in poor and developing countries. To date the GAVI has helped immunize over 822 million children, preventing over 14 million deaths worldwide. More than 1.1 billion vaccines have been administered over the period of 2000–2019. The GAVI also helped increase DPT (DTP3) vaccine coverage in supported countries from 59% in 2000 to 81% in 2019, contributing to cutting the child mortality rate of 10% in half.

Most virus vaccines have been developed in the second half of the 20th century, boosted by the discovery of cell and virus cultures by John Enders [12]. Poliovirus was targeted by the WHO because it is a vaccine-preventable infection caused by a non-zoonotic enterovirus (with 3 serotypes PV-1, PV-2 and PV-3). The infection causes flu-like symptoms, evolving after a short phase of improvement ranging from no symptoms to febrile, to a flaccid and asymmetrical muscular paralysis of the limbs, respiratory muscles, or both [14]. Since paralysis is caused by motor neuron death caused by viral replication, it is irreversible and leads to denervation of muscular atrophy.

There are two types of vaccines, an injectable polio vaccine (IPV) developed by Jonas Salk in 1955, and an oral polio vaccine (OPV) developed in 1960 by Albert Sabin [15]. The vaccine discovered by Jonas Salk was evaluated in the largest medical trial ever held, led by Thomas Francis. The clinical trial eventually included 1.8 million children in 44 states, from Maine to California. The results of the field trial were announced on 12 April 1955 the tenth anniversary of the death of President Franklin D. Roosevelt, arguably the most famous polio patient. The Salk vaccine was 70% effective against PV-1 and over 90% effective against PV-2 and PV-3. Many people rushed to get vaccinated all over the word. However, the mass vaccination campaign was sadly affected by an industrial incident known as the Cutter incident. In April 1955, more than 200,000 children in five Western USA states received a polio vaccine in which the process of inactivating the live virus proved to be defective. Subsequent investigations revealed that the vaccine, manufactured by the California-based family firm of Cutter Laboratories, had caused 40,000 cases of polio, leaving 200 children with varying degrees of paralysis and killing 10. 

Unlike the IPV, the OPV induces good intestinal immunity, is more effective at blocking transmission, is cheaper, and does not require highly qualified personnel for administration. Thus, the WHO has implemented OPV for vaccination campaigns in developing countries. Before 1950 and the availability of vaccines against polio, the natural poliovirus infection caused about 600,000 deaths annually. Since 1988 and the beginning of the WHO eradication campaign named the Global Polio Eradication Initiative (GPEI), the number of cases has dropped by more than 99.99%: in 1988 there were 35,000 reported cases and 155 endemic countries; in 1991, 12,247 reported cases and 130 endemic countries; 1994, 8500 reported cases and 60 endemic countries; 1997, 5185 reported cases and 42 endemic countries; 2000, 2971 cases and 20 endemic countries; 2003, 784 cases and 7 endemic countries; and in 2006, 1997 cases and 4 endemic countries. By 2021 only two countries recorded cases: Afghanistan and Pakistan. There were 22 cases of wild PV-1 infection in 2017, while 33 cases were recorded by the WHO in 2018. Although the annual rate of cases is very low, the finish line of reaching eradication seems very hard to cross.

As of late 2019, the WHO has declared that among the three wild poliovirus (WPV) serotypes, only type 1 (PV-1) is circulating [16,17]. Although annual cases are very low, reaching complete polio eradication has become challenging. The challenge comes from the emergence of circulating vaccine-derived poliovirus (cVDPV), which reacquired neurovirulence; provoking vaccine-associated paralytic poliomyelitis (VAPP) [3]. More than one hundred cases of VAPP were recorded between January 2017 and June 2018 in at least five different countries [16]. Vaccine-induced polio cases far exceed natural cases [3,18]. In 2019, 280 cVDPV cases were recorded as compared to 165 wild PV cases [17]. Simply stated, the main cause of polio is its vaccine [3]. In April of 2016, the trivalent OPV vaccine was withdrawn and replaced by a divalent OPV vaccine without serotype 2 (the leading cause of VAPP). For polio, we are now in the critical period where meticulous surveillance of the disease epidemiology is required for success [19]. The case of poliovirus is a particularly striking example of the unforeseen complications and increased public health efforts that must be invested in to eradicate an endemic disease. To make matters worse, poliovirus surveillance data of 2019 indicated an increase of wild PV-related cases [20], suggesting the end is further than expected. The public’s view of the WHO polio campaign is generally positive; though the thought of using synthetic biology to build a better vaccine is perhaps less so.

## 3. Reverse Genetics for Rational Vaccine Design

The first use of reverse genetics in human diseases started in 1981 with poliovirus was produced from a cDNA clone [21]. This launched the reverse genetics approaches still used today for isolating and recovering many positive-strand RNA viruses. The extension of reverse genetics to negative-strand RNA viruses was more laborious as genomes from these viruses are not directly ‘infectious’. The first negative-sense RNA virus produced from cDNA was carried out with rabies virus in 1994 [22]. The technique used cDNA of the negative-sense genomic RNA, trans-complemented with plasmids expressing the replication machinery subunits. Large genome positive strand RNA viruses, including the large coronavirus family, were too difficult to replicate up until 2000 [23,24], with the advent of the bacterial artificial chromosome [25]. Now, the tools of reverse genetics are widely used. They facilitate virus exchange between scientists through the exchange of stable DNA plasmids, attenuated pathogenic viruses, and allow for rapid response to outbreaks of novel virus strains in recurring diseases such as influenza virus [25].

Even before SARS-CoV-2’s emergence, most of the vaccines arriving on the market were developed through reverse genetics at some point, such as human papillomavirus, dengue virus, and Ebola virus [26,27,28]. Notably, virus-like particles (VLPs), which rely on reverse genetics, have flourished after the market success of the human papillomavirus vaccines (Gardasil^®^, Gardasil 9^®^, Cervarix^®^) [26]. Another example includes a vaccine against dengue (Dengvaxia^®^) using a chimeric virus of dengue envelope proteins in the Yellow Fever vaccine backbone (YFV 17D204) [27]. Even more recently, a recombinant vaccine against Ebola (Ervebo^®^) has been authorized in Europe and in the USA after a successful evaluation in Guinea [29]. Ervebo^®^ was originally constructed by reverse genetics, the glycoprotein of the vesicular stomatitis virus being replaced with the glycoprotein of Ebola virus [28]. The ongoing vaccine development using these tools have built a foundation on which the public has accepted (implicitly or otherwise) the risks for the rewards of reverse genetics.

## 4. Synthetic Biology: Two Steps Forward

Synthetic biology was born by borrowing some tools from the world of engineering. Its purpose is to artificially design and create biological elements from various organisms to exploit their function in novel ways. In 2002, synthetic biology was used to recreate poliomyelitis virus from synthetic cDNA sequences [5]. In 2005, the synthesis of the 1918 pandemic influenza virus was published [11]. Justification for its synthesis revolved around expanding knowledge as well as using it as a target for testing drugs and vaccines. This has been criticized due to the risk of possible escape [30].

In 2017, *Science* announced the circulation of a manuscript describing the synthesis of horsepox virus, which disappeared in the 70′s. The synthesis was published in January 2018 [6]. The rationale for the approach was to produce a smallpox vaccine. The way in which the synthesis happened so matter-of-factly, as a 200,000 base-pair genome mail-ordered synthesis, did not placate concerns about the relevance of the research. The publication of the horsepox virus synthesis received severe scientific criticism by several journals [7,31,32]. While the reasoning for the synthesis is noble and the research inevitable, the fear tends to mask the potential positive purposes of the technology [33]. It is clear this work highlights the fact that no disease-causing organism can no longer be fully eradicated [7].

There are many types of technology that fall under synthetic biology. Currently, there are vaccines that are DNA-based [34], RNA-based [35]; ones that use viral vectors [36,37], artificial liposomes/liposomal nanoparticles [38,39], or are attenuated by computational codon design [40,41]. The most popular, in terms of visibility to the layperson, is the use of nucleotides (specifically RNA) as the vector for vaccination. The Pfizer and Moderna SARS-CoV-2 vaccines are both based on this technology [42]. Briefly, a short piece of RNA is synthesized and suspended in an artificial vacuole that is injected intramuscularly to transfect muscle cells to produce and secrete a peptide analogous to part of a viral protein (in this case, a piece of the spike protein). The advantages are clear; a rapidly deployable and modifiable vaccine that can respond to fast-changing epidemics [43]. Self-replicating RNA/DNA vaccines have an added advantage of increasing the protein production in the transfected cell, which is an important aspect of this type of vaccine development [44,45]. Supplied directly in frame or through trans-expression of a replicase cartridge, typically the alphavirus NSP1-NSP4 series of non-structural replication components, and the proper Cap and 5′- and 3′-untranslated regions (UTRs), the injected RNA and transcribed RNA from a DNA vaccine are able to replicate as the genome of an alphavirus would; entirely in the cytoplasm and with high abundance. The result is as would be expected, an increased protein production, and hopefully a greater immune response against the antigen generated [44,46,47]. For DNA-based vaccines, other mechanisms of delivery include gold nanoparticles adsorbed with DNA that are then injected by a gun [48,49], naked DNA injection, and application on mucosal membranes [50,51,52]. The amount of nucleotide required for each method varies greatly, from nanograms in the case of adsorbed DNA on gold particles, to hundreds of micrograms for naked DNA and mucosal applications. Despite the unequal efficiency in delivering the nucleotides, each mechanism has its benefits in targeting specific organs, preventing damage due to the vaccination method, or creating a specific immune response, such as an IgA response on a mucosal membrane [52].

Taking the next step, codon manipulation, and completely synthetic virus genomes can take advantage of a virus’ innate targeting ability and replication machinery to vaccinate the host. In fact, this technology has resulted in attenuation in poliovirus, Influenza A virus (H1N1), Respiratory syncytial virus, arenaviruses, and others [53,54,55,56,57,58,59,60]. Codon manipulation takes advantage of the natural bias present in the frequency and base-pair preferences of mammalian cells when compared to a predicted normal distribution and to other species. By including underrepresented codons into the genome, translation efficiency is reduced, as there are fewer precise tRNA present to accommodate the unnatural balance of codons, or take advantage of the third-position wobble pairs [61,62,63,64]. This can be extended to codon pairs (dinucleotides), by decoupling codons that are commonly seen together. The reasoning behind this is somewhat unclear; however it likely results from inefficiencies from both uncommon codons and an overrepresentation of CpG and UpA dinucleotides [59,65,66]. This is important, as there is an extreme bias against CpG dinucleotides in the human genomes and the underrepresentation of UpA/TpA across all species. These are specific targets of exploitation that can be used to further attenuate a virus without creating selective pressure through protein modification. The reason for the bias is unclear [59,66,67,68,69,70], but could be due to a natural migration due to methyltransferases [66,71] or simply a result of some additional fitness from the revised sequence that was selected; the latter being supported by the associated tRNA expression levels [59,72,73]. Regardless of the reasoning, the bias has resulted in immune monitoring proteins that respond to the underrepresented dinucleotides. ZAP, the Zinc-finger antiviral protein was initially identified as an inhibitor of Moloney murine leukemia virus [74,75,76,77] and is now associated with a number of other viral inhibition mechanisms [74,76,78,79,80,81]. Structural analysis of ZAP and an RNA ligand show its preference for CpG sequences [82], and a CpG enriched HIV-1 was released from ZAP antiviral activity through mutation surrounding the CpG binding pocket [83]. Another protein, RNAse L, has been implicated in monitoring UpA and UpU dinucleotides, and some viruses, especially those with a high UpA concentration such as hepatitis C virus, have a particular sensitivity to RNAse L activity [84,85]. By inserting an increasing number of CpG and UpA into the genome of a vaccine candidate, the likelihood attenuation can be achieved through hydrolysis of mRNA from the virus or potential viral genomic RNA in the case of RNA viruses. Another avenue of research with respect to codon manipulation is controlling the evolutionary platform the virus has to work with [40,86,87]. This is most commonly accomplished by introducing synonymous mutations in serine and leucine that make it 1-step away from mutating into a stop codon. This restricts the evolution of the genome by making it difficult to alter certain aspects of a protein without completely eliminating its function. Of course, there is the potential to drive viral evolution in direction beneficial to the virus [88,89]. As an emerging field, synthetic biology, especially as it relates to viruses, is still in its infancy despite the wealth of knowledge it has already yielded about genomic manipulations and synthetic viruses. There is hope that genetic manipulation on both a pre- and post-transcribed level will give us the tools to finally capture and control the elusive infectious diseases that spread through the population.

## 5. Synthetic Biology: One Step Back

With advancement into a new field of vaccine research comes stumbling blocks and hurdles that must be overcome or mitigated in order to have a successful product. Currently, the disadvantages of nucleotide-based vaccines have begun to emerge in the public’s eye that speak to more challenges that exist behind the curtain. As has been implied by the low/ultra-low temperature storage for these vaccines, the synthesis and stability of both the RNA molecule, and the composition of the lipids making the nanoparticles are variables that are getting the real-world pressure they need for testing and improvement [42,43,90]. The concept has been around since at least the early 90′s [91,92]. In addition, the expression of protein from the nucleotide-based vaccines is important to ensure an adequate immune response. The drawbacks of nucleotide-based vaccines are somewhat paralleled between DNA and RNA technology. Stability of the nucleotide is less problematic with DNA, naturally, however the introduction of dsDNA, ssRNA, and/or dsRNA into the cytoplasm can result in Type I Interferon production [93]. While normally beneficial against a viral infection, expression of Type I interferons can reduce the expression of exogenous proteins [94,95,96], and can suppress the immune response of B-cells and macrophages [97]. There is an additional complication, of which codon manipulation takes advantage of, where the injected sequence is likely to have an increase in CpG, and UpA dinucleotides. Designing the mRNA to include modified nucleotides, such as pseudouridine and its derivatives, can greatly reduce the antigenic nature of the nucleotide-based vaccine [98,99]. Of course, this does not help a self-replicating RNA construct, which would ideally have to outperform the hydrolysis from the cells’ immune responses.

The matter of getting the nucleotide into the cells is a matter of discussion as well. Traditional viral vectors can transfect a target cell somewhat readily, though there is an undesired immune response that can arise against the vector itself, where adenoviruses have typically been used as vectors for vaccines [100,101]. There are other vectors including rhabdoviruses, poxviruses, alphaviruses, and adeno-associated virus [37,102,103,104,105]. They all have some advantages and disadvantages, including possible dual-use from the knowledge gained by manipulating virulence [106]. Naturally, non-viral vector methods largely avoid these disadvantages, but come with a list of their own including nucleotide stability, carrier cytotoxicity (especially with the novel lipids in nanoparticles), and efficient protein expression. While solutions are forthcoming, it will take time to determine the extent to which this new technology can be used to drive vaccine synthesis. If the hurdles are overcome, is there potential for a single-shot RNA-based vaccine that delivers long-lasting immunity?

## 6. Toward a Rapid Vaccine Response

During the 2009 H1N1 influenza outbreak, despite delays, the pharmaceutical industry ran the first race for global vaccine development in history. More recent data have shown that isolation of influenza vaccine clones by reverse genetics can be synthetically produced within 5 day of sequence release [107]. Two recent studies have proven that reverse genetics and synthetic biology were readily available after SARS-CoV-2 emergence [8,108]. Synthetic biology allowed the production of SARS-CoV-2 virus in less than a month; including 3 weeks for ordering the gene synthesis [8]. The technology behind the rapid synthesis is based on using a yeast artificial chromosome (YAC) produced in *S. cerevisiae* to carry a copy of the viral genome. Recreation of SARS-CoV-2 virus has changed the time-gap between disease discovery and vaccine development. Indeed, other companies, are using fully synthesized genomes to create live-attenuated vaccines [109]. The turn-around time from sequence to manufacturing for clinical trials is on the order of months. While not as rapid as nucleotide-based vaccines, it comes with the advantages of properly folded, whole-virus antigen exposure the immune system, hypothetically creating a broad immune response to all available targets of the virus.

The SARS-CoV-2 RNA-based vaccines are the fastest ever developed and largely produced vaccines, however it is unclear to some why these RNA-based vaccines were successful when other projects focusing on RNA-based vaccines were abandoned. There is evidence that in vitro testing of liposomal-based vaccines are weakly correlated with their success in vivo [110]. There is the possibility that the undesired interferon response that comes from the introduced RNA enhanced the immune response to the expressed protein. It could have been merely a culmination of decades of research that finally yielded a viable product. The real stability of the product is also unknown. The long-term storage temperature and time allotted to keeping it cold once reconstituted is changing [111]. This is not bad news, but merely watching vaccine research progress in view of the public. This is usually completed behind the curtain under the scrutiny of peers and sponsors, and only when the product is fully tested and ready for distribution does the public see the results [112]. It is an exciting time to watch the world piece together the process of testing and evaluating a new vaccine technology. The results of the vaccination movement have been clear though; the technology works [42].

## 7. Conclusions

As with the smallpox and poliovirus eradication campaigns, the answer to virus eradication is not merely dependent on the efficacy of the vaccine used. Despite the COVID vaccination campaign’s likely success, due to the fact that it is (a) new and (b) unprecedentedly fast, there is pushback against accepting the vaccination [113,114]. Hearing about potential breakthrough mutant strains of SARS-CoV-2 and talk of an annual booster against new strains causes additional stress and concern in general. Overall, the safety of these vaccines are still being evaluated and the results are unpublished. Concerns of who gets the vaccine and who must wait, especially those in very rural areas and in developing countries that lack the infrastructure to manage the receipt and distribution of a fragile vaccine, will likely continue to exist as more and more vaccines are developed using RNA and lipid nanoparticles. CureVac has a vaccine candidate going through clinical trials that do not require sub-zero temperatures for storage using similar RNA-based technology to that of Pfizer and Moderna [115,116]. This is an indication that a point will come where distribution and administration of this type of vaccine will become easier in harder to reach locations. That said, it is imperative to use any and all vaccines developed against a viral disease to accelerate protection and defend against novel strains, especially with respect to RNA viruses that tend to have somewhat mutable genomes by nature. 

If the current trend of successful antiviral response holds, synthetic biology will have presented to the public an arsenal of tools for managing viral diseases. The success or failure of the SARS-CoV-2 campaign will have a lasting impression on the public view of synthetic biology. The inherent risks of re-emergence of viral diseases that were eradicated or are in the process of being eradicated (i.e., smallpox, polio), while always present, will merely become a risk that must be mitigated. How will it be mitigated? Should it be regulated? Should sensitive DNA sequence synthesis be restricted to a few administratively approved laboratories? The restrictions put in place will be liken to a padlock on a chest; deterring none but the honest thief. This will only continue to be the case as mail-order synthesis, and synthesizers in general, improve in quality and nucleotide synthesis length. Nevertheless, barriers are being put into place to prevent unwanted abuse of gene synthesis [117].

As the goal of viral disease eradication approaches, the ability to recreate historically eradicated viruses using synthetic biology has the potential to jeopardize the long-term sustainability of eradication. However, the emergence of the SARS-CoV-2 pandemic has highlighted our ability to swiftly and resolutely respond to a potential outbreak. This virus and its vaccines have been synthetized faster than any other in the past and is resulting in vaccines before most attenuated candidates reach clinical trials. COVID-19 currently presents an opportunity in which synthetic biology can clearly facilitate the management and possible eradication of a disease. The perception of this new technology gives a tangible benefit to and improves the public opinion of a previously academic technology. Synthetic biology needs to use this opportunity to demonstrate its truest potential to the public, gain trust in the technology, and solidify a footing in the world of vaccines.

## Figures and Tables

**Figure 1 viruses-13-00569-f001:**
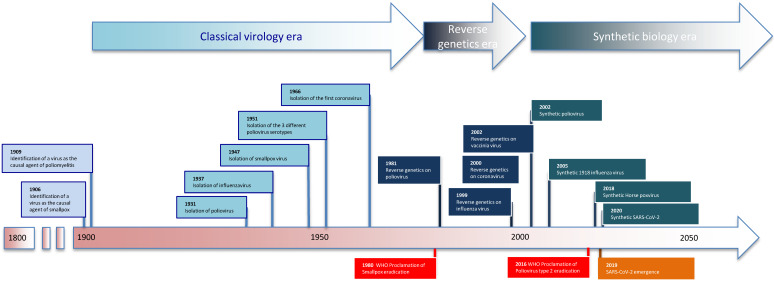
The light blue rectangles are virus discoveries, dark blue rectangles are for reverse genetics-based discoveries, and blue-green rectangles are for virus synthesis. The orange rectangle represents virus emergence, while red is for viral eradication as declared by the World Health Organization (WHO).

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
