# Peer review of "Virus Eradication and Synthetic Biology: Changes with SARS-CoV-2?"

_viruses, 2021, doi:10.3390/v13040569_

Round 1
Reviewer 1 Report
In this study, Jean-Nicolas Tournier and Joseph Kononchik discuss about the use of synthetic biology to create or recreate viruses or as a possible vaccine solution to eradicate one or more viruses.
To discuss this perspective the authors used the following outline:
-Introduction.
In this part, the authors point out that the only virus eradicated, so far, is smallpox (eradication celebrated by the WHO in 1980), but also that the announcements of that time predicted the imminent eradication of malaria, polio and measles. The authors then emphasize that several years after the announcements of the 1980s, the malaria program was discontinued, and that polio eradication has still not been achieved (outbreaks of circulating vaccine-derived poliovirus). Finally, the authors briefly discuss the positive and negative aspects of the synthetic biology.
- The challenges of virus eradication.
In this part, firstly, the authors briefly review the history of smallpox eradication. Second, they focus on the history of oral poliovirus immunization by noting that today there are still cases of contamination by the wild PV but also cases related to the circulation of vaccine-derived poliovirus.
- Reverse genetics for rational vaccine design.
In this paragraph, the authors, first of all, discuss about the success of the use of reverse genetic for vaccine design. Then, in a second time, the authors debate of the emergence of synthetic biology and its application to create or recreate viruses (PV, 1918 influenza virus, horsepox virus) emphasizing that this technology allows at any time to possibly recreate disappeared or eradicated organisms.
- Toward a rapid vaccine response.
In this last part, the authors focus on the ongoing SARS-CoV-2 pandemic and highlight the role of synthetic biology in the very rapid vaccine development.
The authors conclude that the use of synthetic biology has been a major contributor to the development of a SARS-CoV-2 vaccine in unprecedented time.
This article asks a very interesting question about the use of synthetic biology in the field of viruses, goes back over the history of vaccines and the problem of eradicating viruses. However, from my point of view, the manuscript hovers the subject and still needs improvement to reach its goal, as numerous articles on SARS-CoV-2 and vaccine (and the use of synthetic biology) have already been published. From my point of view, I think that this manuscript should be enriched and should not be submitted as a communication but as a review.
Major comments
#1 Section: "Challenges to eradicating the virus".
It would be good to introduce the Alma-Ata Conference (1974): "Extend" smallpox vaccination to six killer diseases in early childhood with an effective vaccine (tuberculosis, tetanus, diphtheria, pertussis, polio, and measles).
Moreover, I think it's essential to go back over the history of poliovirus vaccine as a whole by introducing the work of Jonas Salk and explaining the design of his inactivated vaccine. Similarly, I think it is also important to recall the history of the U.S. vaccination campaign based on this vaccine (benefits and incidents). Indeed, in 1955, cases of polio were discovered in children who had just been vaccinated. Eleven of them died and some children were paralyzed. The vaccination campaign was suspended until it was discovered that in some vaccine lots the virus had not been sufficiently inactivated. In the same vein, I think it would be good to go into more detail about the history of the attenuated vaccine developed by Albert Sabin (how the vaccine was produced) and why his vaccine has been widely used in developing countries in particular (lower cost of production and easier administration).
I also think it would be appropriate to give numbers of polio cases published by the WHO over the last few years (1988, 35,000 reported cases and 155 endemic countries; 1991, 12,247 reported cases and 130 endemic countries; 1994, 8,500 reported cases and 60 endemic countries; 1997, 5,185 reported cases and 42 endemic countries; 2000, 2,971 cases and 20 endemic countries; 2003, 784 cases and 7 endemic countries; 2006, 1997 cases and 4 endemic countries etc.), to emphasize on the contribution of both oral and inactivated polio vaccines ( describe the Global Polio Eradication initiative (GPEI) that has reduced the global incidence of wild type poliovirus such that serotypes 2 and 3 have been declared eradicated (WHO. Two out of three wild poliovirus strains eradicated - Global eradication of wild poliovirus type 3 declared on World Polio Day, https:// www.who.int/news-room/feature-stories/detail/two-out-of-three-wild-poliovirusstrains- eradicated (2019)). I think it would be also interesting to comment on the resurgence of polio in certain countries such as Nigeria, Afghanistan, India and Pakistan (Pakistan: 6 vaccinators murdered December 2012, Nigeria: Attack of 2 hospitals performing polio vaccination March 2013). Finally, it will be nice to compare briefly the advantages and disadvantages of whole inactivated viruses to attenuated viruses.
#2 After the “Reverse genetics for rational vaccine design” which needs to be more detailed.
It would be important to develop a specific third section to synthetic biology. In this section, it will be essential to describe codon deoptimized vaccines that have already been developed for poliovirus (but also H1N1 etc.) and to discuss the use of codon deoptimization as an attenuation technique (advantages and disadvantages: increase in pro-inflammatory dinucleotides such as CpG and UpA). In addition to these aspects, vaccines based on DNA or RNA (advantages and disadvantages) need also to be describe in details. Indeed, the synthetic biology (easy synthesis of nucleic acids for rapid and effective vaccine design) is just the tip of the iceberg. One of the great challenges for RNA-based vaccines for example, beyond easy synthesis, has been to provide solutions to increase their intracellular stability, reduce cytotoxicity and improve protein expression (RNA structure: addition of a Cap, of an untranslated region, base composition: pseudouridine, methoxyuridine etc, self-amplifying mRNAs). In this section the paragraph about synthetic viruses will have to be added as well as the section on SARS-CoV-2.
#3 Finally in a last section, the authors should discuss on the use of synthetic biology in the vaccine application of today and tomorrow and also the use of synthetic biology to create or recreate vaccines (should such practices be regulated? etc). The authors should also debate the following major points:
- What about access to synthetic vaccines for all countries (constraints of Pfizer and Moderna vaccines due to their storage at -80°C, a major issue for developing countries that do not have the necessary infrastructure).
- Eradication of a virus is not only related to the performance of a vaccine, issues such as acceptability and ease of administration are equally important (sociological, cultural, economic issues, etc.).
#4 Figure. Most of the text in the figure is impossible to read. The figure needs to be improved and also completed.
Reviewer 2 Report
The manuscript is a solid perspective of potential of synthetic Biology and applications /implications during the SARS-CoV-2 pandemic. The introduction guides to the past achievements of vaccination, the subsistent problems and solutions, as well the "dual usage" of synthetic genes/virus.
The authors in a clever way pointed to the advantages of reverse genetics towards a rapid vaccine response, highlighting all the successes of this approach but confronting with the opportunity and risk of being used to recreate ancient virus and pandemics jeopardizing all the efforts to eradicate extinct threads. The conclusion could be more substantiated, with a clear reinforcement of the role of synthetic biology on this pandemic.
Minor corrections: The figure is difficult to read, I would suggest redoing to improve legibility of the figure.
Overall, I believe this is an important perspective and should be published with minor revisions
Author Response
Point 1:
We will work with the journal to improve the resolution of the final figure.
Round 2
Reviewer 1 Report
Dear Editor,
The manuscript has been considerably improved and now warrants publication in Viruses
There is one minor point remaining, which concerns the quality of the figure.
Indeed, most of the text in this figure cannot be read. The figure needs to be improved.
Sincerely yours,
Nathalie Chazal